# A Recovery-Oriented Program for People with Bipolar Disorder through Virtual Reality-Based Cognitive Remediation: Results of a Feasibility Randomized Clinical Trial

**DOI:** 10.3390/jcm12062142

**Published:** 2023-03-09

**Authors:** Alessandra Perra, Alessia Galetti, Rosanna Zaccheddu, Aurora Locci, Federica Piludu, Antonio Preti, Diego Primavera, Lorenzo Di Natale, Antonio Egidio Nardi, Peter Konstantin Kurotshka, Giulia Cossu, Federica Sancassiani, Giusy Stella, Valerio De Lorenzo, Thurayya Zreik, Mauro Giovanni Carta

**Affiliations:** 1International PhD in Innovation Sciences and Technologies, University of Cagliari, 09124 Cagliari, Italy; 2Department of Medical Sciences and Public Health, University of Cagliari, 09124 Cagliari, Italy; 3Department of Neuroscience, University of Turin, 10125 Turin, Italy; 4IDEGO Digital Psychology Society, 00133 Rome, Italy; 5Institute of Psychiatry, Federal University of Rio de Janeiro, Rio de Janeiro 21941-901, Brazil; 6Department of General Practice, University Hospital Wuerburg, 12459 Wuerzburg, Germany; 7Department of Mental Health and Pathological Addiction, ASL 5, 00034 Rome, Italy; 8PRoMIND Services for Mental Health, 00133 Rome, Italy; 9Mental Health Service User Association, 11072070 Beirut, Lebanon

**Keywords:** virtual reality, Cognitive Remediation, mental health, recovery

## Abstract

Background: Cognitive impairment is a frequent consequence of bipolar disorder (BD) that is difficult to prevent and treat. In addition, the quality of the preliminary evidence on the treatment of BD through Cognitive Remediation (CR) with traditional methods is poor. This study aims to evaluate the feasibility of a CR intervention with fully immersive Virtual Reality (VR) as an additional treatment for BD and offers preliminary data on its efficacy. Methods: Feasibility randomized controlled cross-over clinical study, with experimental condition lasting three months, crossed between two groups. Experimental condition: CR fully immersive VR recovery-oriented program plus conventional care; Control condition: conventional care. The control group began the experimental condition after a three months period of conventional care (waiting list). After the randomization of 50 people with BD diagnosis, the final sample consists of 39 participants in the experimental condition and 25 in the control condition because of dropouts. Results: Acceptability and tolerability of the intervention were good. Compared to the waitlist group, the experimental group reported a significant improvement regarding cognitive functions (memory: *p* = 0.003; attention: *p* = 0.002, verbal fluency: *p* = 0.010, executive function: *p* = 0.003), depressive symptoms (*p* = 0.030), emotional awareness (*p* = 0.007) and biological rhythms (*p* = 0.029). Conclusions: The results are preliminary and cannot be considered exhaustive due to the small sample size. However, the evidence of efficacy, together with the good acceptability of the intervention, is of interest. These results suggest the need to conduct studies with larger samples that can confirm this data. Trial registration: ClinicalTrialsgov NCT05070065, registered in September 2021

## 1. Background

Cognitive impairment (CI) is associated with social and functional impairment in individuals that suffer from mental health disorders [1,2]. CI could be defined as a complex relationship of selective hypo and hyperactivity networks linked to attention, verbal fluency, memory, and executive function [2,3]. CI is an important target for rehabilitation for people with bipolar disorder (BD) [3,4,5]. BD is a common, chronic disorder and one of the leading causes of disability worldwide [6,7,8,9]. It is associated with a frequent neurocognitive impairment during the manic, depressive and euthymia phases [10,11], with a long-term risk of developing dementia [12,13]. People with BD often display neuropsychological deficits in their attentional capacities and executive functions (flexibility, working memory, and inhibitory control) and processing speed and verbal/memory learning [13,14,15]. These CI aspects are a barrier to achieving clinical, personal, and social improvements that are essential for a good quality of life [16,17,18]. CI is highly prevalent in patients with BD. About 30% were impaired in at least two different cognitive domains [17]. Although psychopharmacological treatments associated with psychosocial interventions [19,20,21] contribute to the improvement of some core symptoms of BD, such as depressive/manic symptoms, cognitive deficits do not improve and get worse over time [10]. Except for the indirect positive lithium effect, most treatments have cognitive side effects caused by their extrapyramidal, sedative, anticholinergic and blunting mechanisms. Furthermore, given the alteration of the circadian rhythm (including social and behavioral rhythms) that occurs with the onset of BD [7,22] and the comorbidity of anxiety symptoms [23,24], cognitive rehabilitation should be an aim of treatment in BD, in order to promote recovery and a human rights-based approach [25]. Different psychiatric rehabilitation interventions of proven effectiveness, such as Cognitive Remediation (CR) programs and physical activity, are currently used to reduce the cognitive and clinical impairment of people with mental and neurodegenerative disorders [26,27,28,29,30]. These interventions also play a role in the primary prevention of cognitive decline and alterations of the social and behavioral rhythms in healthy populations [31,32,33,34,35]. CR programs include behavioral training aiming to improve cognitive functions, social cognition and metacognition, and the generalization of the achieved goals in daily life [36]. They are effective in the treatment of different mental and neurodegenerative disorders [37,38,39,40,41,42,43,44]. In addition to traditional approaches, different methods of CR interventions (computerized, paper and pencil, in individual or group settings) are available [43,45,46]. Preliminary evidence for the use of traditional CR methods for BD is still limited due to the high risk of bias associated with the sample size and dropout rates [47]. In line with the WHO innovation objective in mental health and global digitalization [48], the use of technologies for the assessment and treatment of mental and neurodegenerative disorders is increasing [49,50]. Virtual Reality (VR), which engages people in playful and ecological scenarios and facilitates the generalization of the trained skills [47,51,52,53,54], is considered an effective add-on intervention in psychosocial rehabilitation [50], particularly in social cognition training (e.g., social/occupational skill training in people with schizophrenia or autism) [50,55] and in psychotherapy to treat anxiety, phobia and post-traumatic syndrome disorders [56,57,58]. VR has also proven to be effective in assessing and treating cognitive deficits in people with Mild Cognitive Impairment (MCI) and Alzheimer’s disease and dementia [59,60,61,62]. The use of fully immersive VR-based CR programs is increasing [63], particularly in the treatment of MCI and schizophrenia. To date, the methodological quality of preliminary evidence is poor [64,65,66]. To our knowledge, no studies on the use of fully immersive VR as a CR intervention aiming to improve the cognitive, personal and social functioning of people with BD are available. Our hypothesis is that a fully immersive VR-based CR intervention could be feasible and clinically effective in people that have experienced BD.

## 2. Aims

### 2.1. Primary Aim

To assess the feasibility of a confirmatory trial that evaluates the effectiveness of the use of a VR tool for the treatment of CI among individuals with bipolar disorder.

### 2.2. Secondary Aim

To evaluate the preliminary efficacy of the trial in terms of intervention’s safety, participants’ satisfaction, and clinical outcomes (regarding cognitive functions and personal and social functioning).

## 3. Methods 

### 3.1. Study Design

This study is a randomized-controlled (two-arm) cross-over clinical feasibility trial. This study follows the reporting guidelines according to the CONSORT extension for feasibility studies [67]. After the experimental group (A) received the VR-based CR intervention as an add-on to conventional treatment and the control group (B) received only conventional treatment for three months (waiting list), group A underwent a one-month interval period which was followed by a phase of conventional treatment, whereas group B received the VR-based CR intervention as an add-on to conventional treatment and became the experimental group.

The trial [63] was registered in ClinicalTrialsgov (NCT05070065, September 2021).

### 3.2. Participants

This trial’s target population included people with bipolar disorder recruited at the Consultation and Psychosomatic Psychiatry Center of the University Hospital of Cagliari (San Giovanni di Dio Civil Hospital), who met the following inclusion criteria: (1) age ranging from 18 to 75; (2) diagnosis of bipolar disorder according to DSM-IV [68]; (3) both sexes. The participants that met these inclusion criteria or their guardians were provided with an informed consent form and signed it before the intervention began. Subjects that did not meet the inclusion criteria, or those who showed a current manic/depressive episode, a diagnosis of epilepsy, or serious eye diseases, were excluded due to the risk associated with the excessive stimulation of virtual reality.

After the randomization of 50 people who met the inclusion criteria, the final sample was composed of 39 subjects in the experimental group (due to some dropouts at the follow-up) and 25 in the control group. Of the participants, 33.3% were males, and 66.7% were females. The mean age in the total sample was 47.23 ± 13.37 (Table 1).

Even if in the registered trial (ClinicalTrialsgov, NCT05070065) we planned to recruit and randomize 60 subjects who met the inclusion criteria, we stopped the enrollment phase early due to the COVID-19 pandemic, which did not allow us to easily involve all the subjects we prevent in the study protocol.

### 3.3. Randomization 

Eligible participants were randomized into two groups. The random allocation sequence was generated using a computer-generated randomization list at the University of Cagliari. Randomization was carried out by a biometrician who was not aware of the participants’ identities and was not involved either in the assessment or in the analysis process.

### 3.4. Blinding

Neither the participants nor the mental health workers of the project could be blinded due to the nature of the intervention.

### 3.5. Intervention

The experimental group was involved in a fully immersive VR-based CR recovery-oriented program. We used the “CEREBRUM” software, one of the most recent VR-implemented CR tools in psychiatric rehabilitation, conceived and designed by “PRoMIND-Services for mental health Srls” (Rome) in association with “IDEGO-Virtual Psychology” (Rome). CEREBRUM is a fully immersive Virtual Reality software created by clinicians and experts specialized in cognitive rehabilitation (psychiatric rehabilitation technicians and psychologists).

It is compatible with the “Oculus Go” virtual reality viewer, a CE-marked device developed by Facebook Technologies in partnership with Qualcomm and Xiaomi that was the hardware of this technology for the present study. The “Oculus Go” is an all-in-one headset containing all the components to provide virtual reality experiences and does not need to be tethered to an external device to use.

The CEREBRUM App allows users to immerse themselves in virtual scenarios that simulate everyday reality, home and urban scenarios (Appendix A). It offers 52 exercises of varying difficulty: 22 exercises are part of the Memory and Learning Module, 10 exercises are part of the Cognitive Estimates Module, and 20 exercises are part of the Attention and Working Memory Module. During the VR exposure, while exploring the 360° scenario, the participants, who could not directly interact with the virtual environments, answered the health worker’s questions. The increasing degrees of difficulty allowed the clinician to adapt the intervention to the participants’ functional diagnosis and to their residual abilities, creating a stimulating learning context in which the exercises were neither too easy nor too difficult. The intervention consisted of 24 sessions of 45 min, divided into two sessions per week over three months. Each session was structured as follows: ➢Reception, psychoeducation and orientation to the tool; ➢Exercise psychoeducation; ➢Psychoeducation to the function to be learned during the exercise; ➢Generalization phase, in which the function and its importance were explained in the participants’ life context (a bio-psycho-socio-cultural approach based on cognition); ➢Execution of the exercise in VR with positive and corrective feedback; ➢Post-exercise comment; ➢Second exercise that used the same method mentioned above (the maximum duration of the exposure to Virtual Reality was 15–20 min); ➢Final comment; ➢Homework, intended as practical suggestions to be implemented by the patients in their daily life. Sessions included an Attention and Working Memory exercise plus one Memory/Learning exercise or one Cognitive Estimation exercise. In some sessions, depending on the participant, the session, and the operator’s assessment, an extra exercise of any type could also be done.

A multidisciplinary team (psychiatric rehabilitation technicians, psychologists, and a psychiatrist) was involved in the intervention. According to the new framework for developing complex interventions [69], the methods used to structure the sessions were replicable and allowed for the promotion of a human-centered approach [70], the accomplishment of cognitive outcomes and improvements in clinical and personal functioning, and the achievement of the generalization of the skills trained. Indeed, we selected this app thanks to the heterogeneity of trained domains (in line with the health need in BD) as a result of a person-centered and recovery-oriented rehabilitation intervention [71], in which the participants could enhance their skills and obtain a global improvement in their health and wellbeing.

### 3.6. Control

The control group consisted of patients put on a waiting list that received conventional treatment, consisting of psychiatric consultations with psychopharmacological drugs administration and/or without psychotherapy.

### 3.7. Outcomes

The primary outcomes regard the feasibility of the study in terms of acceptability and tolerability (dropout rate). They were measured, respectively, as the proportion of patients recruited among those considered eligible and as the proportion of patients completing the trial intervention among those included.

Secondary outcomes regarding the efficacy of the trial included: (1) the intervention’s safety (frequencies of adverse events and severe adverse events); (2) patients’ satisfaction; (3) cognitive functions (visuospatial, attention, memory, verbal and semantic fluency, and executive function); (4) personal and social functioning (anxiety and depression symptoms, quality of life, emotional awareness, social functioning, and biological rhythms regulation).

### 3.8. Data Collection

Patients were screened and enrolled at the Consultation and Psychosomatic Psychiatry Center of the University Hospital of Cagliari (San Giovanni di Dio Civil Hospital). Ad hoc data sheets were created to collect sociodemographic data, level of satisfaction with the program and side effects.

Cognitive functions were evaluated by:-Rey Figure Test [72] for the visuospatial function,-Matrix test [73] and Rey’s Words Test [74] for the immediate recall,-Rey’s Words Test Delayed recall [74], Test of the Tale [75], and Backward Digit Span [76,77] for the memory function,-Forward Digit Span [76,77] and Trail Making Test, part A [73,78] for the attention function,-Phonological and Semantic Verbal Fluency Test, both versions [75,79] for the language function,-Digital Symbol Substitution Test [80,81], Trail Making Test, part B [81], Stroop Test [82], Frontal Assessment Battery—FAB [83] and Cognitive Estimates Test (CET), both versions [84,85] for the executive function.

Personal and social functioning were evaluated by:
-SF-12, Short Form Health Survey, 12 items [86], a self-administered questionnaire that investigates the following dimensions of quality of life and wellbeing: vitality, physical function, physical pain, perception of general health, mental health, physical and emotional health, work functioning and social role;-TAS-20, Toronto Alexithymia Scale-20 item [87], a self-administered questionnaire that evaluates the level of emotional awareness;-SAS, Zung Self-Rating Anxiety Scale [88], a self-administered questionnaire that evaluates perceived anxiety levels;-PHQ-9, Patient Health Questionnaire [89], a self-administered questionnaire that evaluates depressive symptoms;-HoNOS, Health of The Nation Outcome Scale [90], a clinical scale to evaluate general, personal and social functioning;-BRIAN, Biological Rhythms Interview of Assessment in Neuropsychiatry [91], a clinical interview consisting of 18 items that investigates four main areas related to the dysregulation of circadian rhythms (sleep, activity, social rhythms and nutrition).

Participants were assessed before the treatment, after the end of the intervention and six and 12 months after the end of the intervention by the same evaluators who were blinded to the groups that they had been assigned to. In the present study, we report findings regarding pre- and post-evaluation.

### 3.9. Data Analyses

The statistical data were analyzed using the software SPSS (version 21). Frequencies (percentages) or mean ± standard deviation were used for descriptive statistics about sociodemographic variables such as “sex” and “age”, as well as about the level of satisfaction with the experimental intervention and the occurrence of side effects.

Chi-square test and one-way ANOVA were used to test the homogeneity between experimental and control groups regarding “sex” and “age” distributions.

A series of repeated-measure ANOVA was performed, one for each of the outcomes considered, to compare means between the intervention and non-intervention groups over time (pre- and post-intervention) with Bonferroni’s correction. The normality assumption of the dependent variables was tested as sphericity (i.e., variances of the differences between all combinations of related groups must be equal) with Mauchly’s test.

### 3.10. Sample Size Considerations

In the registered trial, we planned to recruit and randomize 60 subjects who met the inclusion criteria. However, we stopped the enrollment phase early due to the COVID-19 pandemic, which did not allow us to easily involve all the subjects we planned to in the study protocol. Therefore, 50 people were randomized to assess the feasibility outcomes. To date, an effective methodology in terms of sample size cannot be established yet, as the evidence in this field of research is limited [38,92,93]. Therefore, the aim of this study is to verify the feasibility and preliminary efficacy of a VR tool for the treatment of CI in people with BD.

## 4. Results

### 4.1. Primary Outcome: Feasibility of the Trial (Acceptability and Tolerability)

As shown in the flow diagram (Figure 1), 119 subjects were contacted, with 62 people becoming enrolled and 12 excluded after evaluation (three did not meet the inclusion criteria, and nine declined to participate). Fifty of the participants were randomized: 25 of them were assigned to group A and received the intervention, and 25 participants were assigned to the control group (group B) and were initially put on a waiting list. They later crossed over and received the experimental intervention. Thanks to the cross-over method, we could include 50 subjects in the experimental group and 25 in the control group. Participants who did not complete at least 50% of the sessions were considered dropouts. Out of the 25 participants of group A, 18 completed all the sessions, and 7 dropped out due to work commitments interfering with the schedule of the interventions or the long distance from their residence and the health service. Out of the 25 participants of group B, 21 completed all the sessions after the cross-over, and 4 dropped out for the same reasons as those dropping out of group A. No one dropped out while on the waiting list. Overall, at the end of the intervention, the experimental and the control groups consisted of 39 and 25 participants, respectively (Figure 1). As shown in Table 1, the experimental and control groups were homogeneous regarding the distributions of the variables “sex” and “age”. Regarding tolerability (dropout rate) and acceptability, around 20% of subjects dropped out, and less than 45% of the people that were contacted did not respond or refused to participate in this trial.

### 4.2. Secondary Outcome: Efficacy of the Trial (Satisfaction with the Trial; Adverse Effects; Cognitive Functions; Personal and Social Functioning)

In the experimental group, 48.7% (19/39) of the participants considered the intervention “excellent” regarding experience, whereas 28.2% and 23.1% considered it “great” and “good”, respectively (Table 2).

Some 76.9% (30/39) of participants did not report any side effects during the first fully immersive VR exposure session. The remaining 23.1% reported the following side effects: emptiness/disorientation, nausea/feeling of emptiness, headache, disorientation, dizziness, tremors/nausea/blurred vision/dizziness, nausea, vertigo and sense of unreality (Table 3).

At the end of the intervention, 87.2% of participants did not report any side effects. As shown in Table 4, the remaining 12.8% reported the following side effects: nausea (two participants), daze (two participants) and a feeling of emptiness/unreality (one participant). Overall, most participants did not experience any side effects.

Regarding cognitive functions and personal and social functioning, we tested the differences reported over time (pre- and post-intervention) between the experimental group that received the VR-based CR intervention as an add-on to conventional treatment and the control group that received conventional treatment (waiting list).

As reported in Table 5, there was an overall improvement in cognitive functions after the VR-based CR intervention. Specifically, we found a statistically significant change after the VR-based CR intervention in the experimental group compared to the control group (Table 5) regarding attention (Matrix, *p* = 0.002; Rey’s Words immediate recall, *p* = 0.019), memory (Rey’s Words delayed recall, *p* = 0.003), verbal function (Verbal Semantic Test, *p* = 0.010), and executive function (CET, *p* = 0.003).

As reported in Table 6, overall personal and social functioning improved after the VR-based CR intervention. In particular, we found a statistically significant improvement in the experimental group compared to the control group regarding depressive symptoms (PHQ-9, *p* = 0.30), biological rhythms regulation (BRIAN, *p* = 0.029) and emotional awareness (TAS-20, *p* = 0.007).

## 5. Discussion

This study showed that a fully immersive VR-based CR intervention for people with BD has good acceptability and tolerability (primary outcome), according to previous psychosocial treatments [94,95,96,97]. The participants also declared great satisfaction with the intervention received, and the majority did not have any side effects. Furthermore, individuals in the experimental group showed an overall improvement in all cognitive, personal, and social functioning after the fully immersive VR-based CR intervention compared to the control group. Particularly, we found a statistically significant improvement (*p* < 0.05) regarding attention, memory, verbal function, executive function, depressive symptoms, biological rhythms and emotional awareness.

To our knowledge, no studies on the use of fully immersive VR as a CR intervention aiming to improve the cognitive, personal and social functioning of people with BD are available. It should be noted that higher dropout rates (around 30%) are usually found in traditional CR interventions (without VR) for people with BD, often carried out with small samples [47]. This data could be interpreted considering the behavioral characteristics of this disorder. As BD is associated with a hyperthymic and exploratory temperament [98], traditional methods (i.e., paper and pencil or computerized) are not actually engaging, particularly in an ecological setting. A dropout rate of around 20% is a notable result in the treatment of CI in BD and suggests that the implementation of innovative interventions such as VR-based allows the achievement of the engagement goal in the treatment of CI, as well as improvements in terms of personal and social functioning. Furthermore, the use of the cross-over method is undoubtedly another strength of this study, as it ensures low statistical variance and has the ethical advantage of including all randomized participants in the clinical intervention. Overall, studies that evaluate the efficacy of VR/CR-based programs in other diseases seem to be effective for the improvement of CI. Even fewer studies examined the functional outcomes related to cognitive improvement [26,27,63].

In using a person-centered method, the VR-based CR intervention is an innovative intervention that allows the participants to learn about their personal resources and to develop strategies for their daily life, thanks to the generalization of the improved skills through personal objectives-focused homework. This, in turn, contributes to the achievement of a global impact in improving their clinical, personal, and social functioning. Additionally, professionals should follow a logical framework to develop complex interventions in mental health and ensure that the outcomes are consistent with the skills trained and with functional improvement [69]. Mental health is a fundamental resource that allows people to achieve daily life goals and exercise their role as citizens of a community [98]. In line with the digital era and the WHO innovation objective [48], increasing the use of technologies in psychosocial rehabilitation could better respond to health needs. Despite the poor methodological quality of the trials, the implementation of CR interventions with traditional methods showed preliminary evidence for people with BD [47]. The results of our study are consistent with other preliminary studies on the efficacy of the use of traditional CR in the treatment of BD and suggest that it is important to implement randomized clinical trials with larger samples to evaluate the clinical efficacy of CR interventions with fully immersive VR in order to confirm and extend this data. One limitation of this study must be cited: we had some dropouts, and the final analyses were only on completers.

### Risk and Benefits

The use of fully immersive VR could be associated with different side effects like dizziness, nausea, headache, eye fatigue, reduced limb control, reduced postural control, reduced sense of presence, and the development of inadequate responses to the real world. However, significant side effects are not expected, as the VR tool has already been used in people with psychosocial disabilities without substantial side effects [99,100]. The benefits of VR in terms of satisfaction and ecological learning and the few side effects satisfy the need to implement an innovative rehabilitation approach in mental health.

## 6. Conclusions

To date, there is no evidence of the use of fully immersive VR as a CR intervention tool to train all cognitive domains of people with BD, even if cognitive impairment is a core component of the social and personal functioning of this disorder. Implementing a randomized clinical trial with a reproducible method developed by a multidisciplinary team specialized in the specific health needs of BD patients is a pertinent research goal in the field of psychosocial rehabilitation. The results of this study are preliminary and not exhaustive due to the limited sample size. However, the evidence regarding efficacy, the great acceptability and tolerability of the intervention are of interest and suggest the need to conduct studies with more extensive samples that can confirm this data.

## Figures and Tables

**Figure 1 jcm-12-02142-f001:**
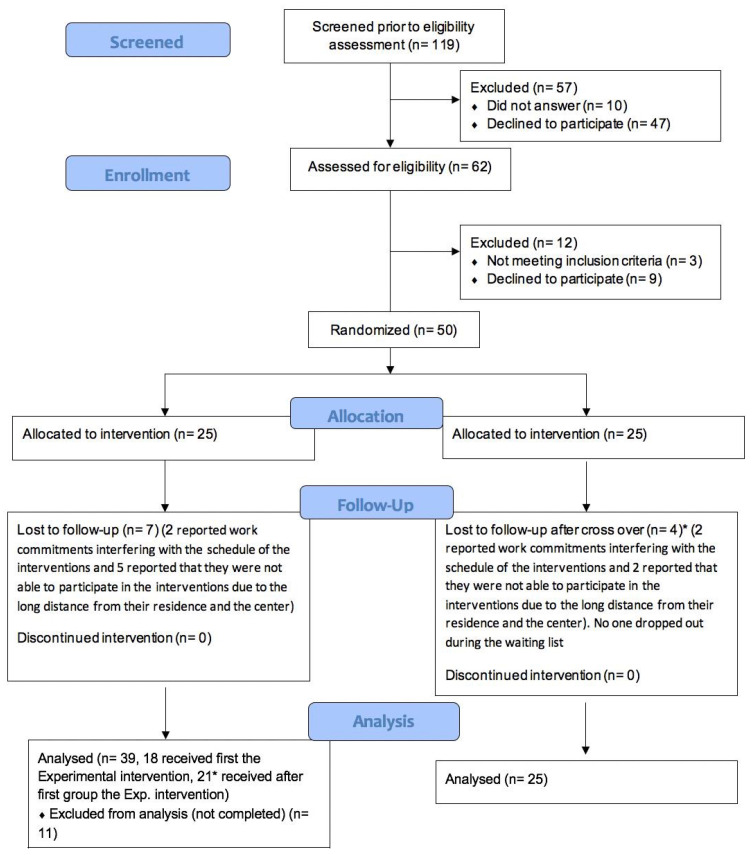
CONSORT flow diagram extension for the feasibility study.

**Table 1 jcm-12-02142-t001:** Baseline demographic characteristics.

		GROUP EXP CONTR		
		EXP	CONTR	TOT
SEX	F N (%)	32 (64)	18 (72)	50 (66.7)
	M N (%)	18 (36)	7 (28)	25 (33.3)
	Chi-Square			0.480
	Sig.			0.606
AGE	N	50	25	75
	Mean ± SD	47.76 ± 13.34	46.16 ± 13.63	47.23 ± 13.37
	F			0.236
	Sig.			0.628

**Table 2 jcm-12-02142-t002:** Frequencies of Satisfaction.

Levels	Counts	% of Total	Cumulative %
3 Good	9	23.1%	23.1%
4 Great	11	28.2%	51.3%
5 Excellent	19	48.7%	100.0%

**Table 3 jcm-12-02142-t003:** Frequencies of Side_Effects_T0.

Levels	Counts	% of Total	Cumulative %
NO	30	76.9%	76.9%
Feeling of Emptiness-Disorientation	1	2.6%	79.5%
Nausea, Feeling of Emptiness	1	2.6%	82.1%
Headache	1	2.6%	84.6%
Disorientation	1	2.6%	87.2%
Dizziness	1	2.6%	89.7%
Tremors, Nausea, Blurred Vision, Dizziness	1	2.6%	92.3%
Dizziness	1	2.6%	94.9%
Vertigo	1	2.6%	97.4%
Sense of Unreality	1	2.6%	100.0%

**Table 4 jcm-12-02142-t004:** Frequencies of Side Effects T1.

Levels		Counts	% of Total	Cumulative %
NO		34	87.2%	87.2%
Nausea		2	5.1%	92.3%
Daze		2	5.1%	97.4%
Feeling of Emptiness	Unreality	1	2.6%	100.0%

**Table 5 jcm-12-02142-t005:** Cognitive Test-Descriptive Analysis (Mean and Standard Deviation): Repeated Measures ANOVA Analyses.

OUTCOMES °	VR/CR GROUP (N = 39)		CONTROL GROUP (N = 25)		*p*		
	Pre	Post	Pre	Post	Time	Group	Time × Group
Figure Rey Immediate (Vis.Sp.)	28.74 ± 8.43	30.92 ± 6.68	28.16 ± 10.20	28.12 ± 8.65	0.002	0.168	0.588
Matrix (Attent.)	1.95 ± 1.38	2.38 ± 1.46	2.52 ± 1.53	2.16± 1.34	0.755	0.617	0.002
Digit Span Forward (Attent.)	2.77 ± 1.53	2.74 ± 1.48	2.84 ± 1.62	2.88 ± 1.48	0.968	0.765	0.853
Rey’s Words Immediate (Attent.)	2.33 ± 1.57	2.95 ± 1.52	2.52 ± 1.55	1.40 ± 1.56	0.109	0.627	0.019
TMT-A (Attent.)	2.87 ± 1.28	3.03 ± 1.42	2.64 ± 1.44	2.96 ± 1.17	0.075	0.527	0.527
Ray’s Words Delayed (Memory)	2.15 ± 1.31	2.77 ± 1.54	2.68 ± 1.31	2.20 ± 1.58	0.707	0.950	0.003
Digit Span Backward (Memory)	1.92 ± 1.62	2.23 ± 1.64	1.84 ± 1.70	2.36 ± 1.68	0.033	0.952	0.579
Test Of Tale (Memory)	2.13 ± 1.39	2.72 ± 1.14	1.84 ± 1.34	2.28 ± 1.34	0.005	0.204	0.675
Verbal Phonological Test (Leng.)	2.64 ± 1.47	3.08 ± 1.26	2.64 ± 1.57	2.84 ± 1.31	0.006	0.729	0.297
Verbal Semantic Test (Leng.)	2.62 ± 1.35	3.23 ± 1.13	2.72 ± 1.51	2.72 ± 1.37	0.010	0.527	0.010
Substit. Digit Symbol (Ex. Fun.)	36.36 ± 14.67	39.04 ± 12.11	37.75 ± 12.19	39.04 ± 12.92	0.302	0.743	0.815
TMT-B (Ex. Fun.)	2.90 ± 1.35	3.05 ± 1.19	2.56 ± 1.32	2.80 ± 1.38	0.103	0.350	0.719
Stroop Test Time (Ex. Fun.)	2.62 ± 1.54	3.03 ± 1.42	2.40 ± 1.68	2.44 ± 1.55	0.204	0.262	0.296
FAB (Ex. Fun.)	15.08 ± 3.12	15.72 ± 2.60	14.64 ± 2.92	15.08 ± 3.61	0.033	0.47	0.686
Cog. Estimation Test (Ex. Fun.)	1.95 ± 1.46	2.77 ± 1.15	2.60 ± 1.19	2.16 ± 1.46	0.346	0.939	0.003

° Sphericity assumption met in all analyses: Mauchly’s test *p* > 0.10.

**Table 6 jcm-12-02142-t006:** Personal and Social functioning: Test-Descriptive Analysis (Mean) Test-Repeated Measures ANOVA Analyses.

OUTCOMES °	VR/CR GROUP (N = 39)		CONTROL GROUP (N = 25)		*p*		
	Pre	Post	Pre	Post	Time	Group	Time × Group
TAS-20	55.00 ± 14.747	49.85 ± 14.982	52.08 ± 14.821	55.76 ± 16.465	0.641	0.675	0.007
BRIAN	49.82 ± 12.380	47.23 ± 11.773	48.12 ± 12.551	50.24 ± 12.387	0.824	0.825	0.029
PHQ-9	13.72 ± 6.121	10.82 ± 6.456	12.20 ± 6.265	11.92 ± 7.455	0.009	0.894	0.030
SF-12	25.95 ± 8.448	28.62 ± 9.193	28.08 ± 7.129	28.48 ± 8.842	0.255	0.566	0.399
SAS	61.05 ± 17.220	53.64 ± 17.609	56.92 ± 15.689	57.96 ± 16.900	0.236	0.978	0.117
HONOS	9.67 ± 6.417	7.38 ± 6.364	10.48 ± 6.771	8.12 ± 6.579	0.006	0.595	0.962

° Sphericity assumption met in all analyses: Mauchly’s test *p* > 0.10.

## Data Availability

Data sharing not applicable.

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
