# Peer review of "A Recovery-Oriented Program for People with Bipolar Disorder through Virtual Reality-Based Cognitive Remediation: Results of a Feasibility Randomized Clinical Trial"

_jcm, 2023, doi:10.3390/jcm12062142_

Round 1

Reviewer 1 Report

This work is relevant because there is currently no culture to evaluate the effectiveness of mental health rehabilitation treatments based on randomized and controlled cross-over clinical studies. Although casuistica does not allow patients in the study universe (patients with bipolar disorder) to receive the results of intervention treatment for TB by repairing cognitive emergence and immersion methods are all stimulants that act as effective in enhancing memory function, attention, cognitive speech fluency, estimation and execution. Evidence also suggests possible success in improving depressive symptoms, signs and biorhythms. It was augurable that the research team continued to deepen these findings, if not expand casuistica to further confirm these results, evidence of effective treatment cognition by repairing the clinical findings in the field that made a significant contribution to the recovery of TB patients.

Author Response

Dear Reviewer,

Thank you very much indeed for the kind consideration and for the positive comments, we really appreciated it.

Reviewer 2 Report

This study on treament of cognitive deficits is addressing an important issue of bipolar disorder that is of much interest for the community. The authors are providing results of a well prepared trial with an ambitious cross-over design on treating cognitive deficits in this field.

Recommendations 

First, overall in the aim to get a more reader friendly version long sentences should be avoided  and paragraphs should be considered.

Second, the description of primary and secondary outcomes in chapter Aims, chapter Results and chapter Discussion should be more unique.

Third, in concern of presentation of results it is confusing to mention demographics in between study results. It is an important question if there are any differences of study groups which has to be answered first. So presentation of results must be extensively revised.

Fourth, the chapter Discussion must be also extensively revised  beginning with a summary of the main results.

Author Response

Dear Reviewer,

Thank you very much indeed for the kind consideration and for allowing us to improve the

manuscript according to the comments of the reviewers.

Our replies to comments are listed below. Changes in the manuscript and the English revision were outlined with yellow color.

Recommendations: First, overall in the aim to get a more reader friendly version long sentences should be avoided and paragraphs should be considered.

Thank you for this observation, we revised the overall presentation of the study aims, distinguishing between primary and secondary aims. Furthermore, we added two paragraphs in the results section to better describe primary and secondary results.

Second, the description of primary and secondary outcomes in chapter Aims, chapter Results and chapter Discussion should be more unique.

Thank you for this observation. As reported above, we fully revised the chapters regarding aims, results, and discussion to make more coherent and unique the entire text.

Third, in concern of presentation of results it is confusing to mention demographics in between study results. It is an important question if there are any differences of study groups which has to be answered first. So presentation of results must be extensively revised.

Thank you for this suggestion. The presentation of the results paragraph has been fully revised, the differences of study groups are not reported firstly because these findings regard the secondary outcomes (effectiveness of the study), we prefer to report firstly findings regarding the feasibility of the study (primary outcome) consistently with the order’s aims.

Fourth, the chapter Discussion must be also extensively revised beginning with a summary of the main results.

Thank you for the comment. We fully revised the discussion and added a summary of the main results.

Reviewer 3 Report

The manuscript entitled “A recovery-oriented program for people with bipolar disorder through Virtual Reality-based Cognitive Remediation: Results of a Feasibility Randomized Clinical Trial” investigates the effect of the use of virtual reality (VR)-based Cognitive Remediation program using “CEREBRUM” software for bipolar patients. This study is novel, since no data has been published to date involving the use of VR paradigm for the treatment of bipolar disorder, although, digitalization accompanies nowadays life and can represent a useful tool for the recovery of mental illness. The study has several advantages. First, the number of used inventories, including those estimating visuospatial, memory, attention, language, and executive function. However, the manuscript requires English checking, there are incorrect sentences.

In addition, there are some major points that have to be clarified and corrected:

1. What is the basis for the use of suggested VR-based CR intervention (“CEREBRUM”)? It would be appropriate to give some explanations in the Introduction. If it is possible, I suggest adding some figures of the examples of the tasks given to bipolar patients within “CEREBRUM” intervention (for instance, as Supplementary data). Also, please indicate the data on the percent of CI accompanying BD in overall in the Introduction.

2. Although the data on the number of men and women comprising the total sample and both groups, as well as mean age±SD is shown in Table 4, it has to be added to Materials and Methods section.

3. It is not stated whether any participants from the Intervention group had a traditional medication during the trial.

4. The authors have to clarify if they have carried out the normality distribution test for the examined cognitive scales; in the case of distribution deviating from the normality it is appropriate to use median and standard error instead of mean and to use non-parametric statistical criteria instead of MANOVA.

5. I suggest to combine Tables 5 and 6 and to drop F(df) for the convenience. The same is applicable to Tables 7 and 8.

6. Since the number of enrolled participants is rather small, the article would benefit from performed power calculations, which can clarify whether the used sample size is appropriate to make statistically significant conclusions.

7. It has to be stated in the Abstract that the final number of involved individuals was 39 in the intervention group and 25 – in the control group, rather than 50 participants, as stated in the Abstract.

There are some minor inconsistencies:

1. Please, reorder the affiliations, since there is no 5th affiliation.

2. The text requires to be checked for the punctuation, and the use of synonyms is required (even in the first sentence). For instance, “individuals” instead of “people” should be used, etc.

3. The authors have to check the appropriate way to present reference numbers in the brackets in the text.

4. The manuscript has to be checked for mistakes (i.e. “Simple size considerations” on p.6).

Overall,  the findings are accurately present and they appear to be valuable.

Author Response

Dear Reviewer,

Thank you very much indeed for the kind consideration and for allowing us to improve the

manuscript according to the comments of the reviewers.

Our replies to comments are listed below. Changes in the manuscript and the English revision were underlined with yellow color.

In addition, there are some major points that have to be clarified and corrected:

  1. What is the basis for the use of suggested VR-based CR intervention (“CEREBRUM”)? It would be appropriate to give some explanations in the Introduction. If it is possible, I suggest adding some figures of the examples of the tasks given to bipolar patients within “CEREBRUM” intervention (for instance, as Supplementary data). Also, please indicate the data on the percent of CI accompanying BD in overall in the Introduction.

Thank you for this consideration and suggestion. We added the prevalence rate of CI in BD and also in the intervention the figures like supplementary material. We didn’t explain in the introduction due to the absence of literature, is a new software and we selected this one for the heterogeneity of the cognitive functions trained and the ecological scenario, in line with the cognitive needs in bipolar disorder.

  1. Although the data on the number of men and women comprising the total sample and both groups, as well as mean age±SD is shown in Table 4, it has to be added to Materials and Methods section.

Thank you so much for the observation, we considered this like a result. We added this information in the participant sub-paragraph in methods section.

  1. It is not stated whether any participants from the Intervention group had a traditional medication during the trial.

Thank you so much for the advice, we better specify that the experimental and control group received standard treatment intervention

  1. The authors have to clarify if they have carried out the normality distribution test for the examined cognitive scales; in the case of distribution deviating from the normality it is appropriate to use median and standard error instead of mean and to use non-parametric statistical criteria instead of MANOVA.

Thank you for this observation that permit us to discover the error. Accidentally, we use the expression “MANOVA” instead of “repeated-measure ANOVA” to describe the statistical analyses we carried out. In this case, the normality assumption is tested as sphericity, i.e., variances of the differences between all combinations of related groups must be equal. Sphericity was tested with Mauchly’s test; in all analyses, the test was not statistically significant. For this reason, we corrected the Methods section.

  1. I suggest to combine Tables 5 and 6 and to drop F(df) for the convenience. The same is applicable to Tables 7 and 8.

            Thank you for the suggestion, we combined 5 and 6 in Table 5, and 7 and 8 in Table 6.

  1. Since the number of enrolled participants is rather small, the article would benefit from performed power calculations, which can clarify whether the used sample size is appropriate to make statistically significant conclusions.

Thank you so much for the consideration. Our study is a feasibility study, it should be calculating the power of the sample size, in line with our results, for the future studies in phase 3.

  1. It has to be stated in the Abstract that the final number of involved individuals was 39 in the intervention group and 25 – in the control group, rather than 50 participants, as stated in the Abstract.

Thank you for the comment, due to the cross-over design we should have had 50 people in the experimental group vs 25 people in the control group, in the results we declared the drop-outs rate of the trial and finally they were 39 in the experimental group.

There are some minor inconsistencies:

  1. Please, reorder the affiliations, since there is no 5th affiliation.

Thank you for the advice, we corrected it.

  1. The text requires to be checked for the punctuation, and the use of synonyms is required (even in the first sentence). For instance, “individuals” instead of “people” should be used, etc.

Thank you for the correction, we improved it.

  1. The authors have to check the appropriate way to present reference numbers in the brackets in the text.

            Thank you, we checked it.

  1. The manuscript has to be checked for mistakes (i.e. “Simple size considerations” on p.6).

Thank you for the advice, we corrected it.

Overall, the findings are accurately present and they appear to be valuable.

Round 2

Reviewer 2 Report

No further comment.